# A Minimal PBPK/PD Model with Expansion-Enhanced Target-Mediated Drug Disposition to Support a First-in-Human Clinical Study Design for a FLT3L-Fc Molecule

**DOI:** 10.3390/pharmaceutics16050660

**Published:** 2024-05-15

**Authors:** Iraj Hosseini, Brett Fleisher, Jennifer Getz, Jérémie Decalf, Mandy Kwong, Meric Ovacik, Travis W. Bainbridge, Christine Moussion, Gautham K. Rao, Kapil Gadkar, Amrita V. Kamath, Saroja Ramanujan

**Affiliations:** Genentech, Inc., South San Francisco, CA 94080, USA; fleisheb@gene.com (B.F.); getz.jennifer@gene.com (J.G.); decalf.jeremie@gene.com (J.D.); kwong.mandy@gene.com (M.K.); ovacik.ayse_meric@gene.com (M.O.); bainbridge.travis@gene.com (T.W.B.); moussion.christine@gene.com (C.M.); rao.gautham@gene.com (G.K.R.); gadkar.kapil.kg1@gene.com (K.G.); kamath.amrita@gene.com (A.V.K.); sarojax@gmail.com (S.R.)

**Keywords:** expansion-enhanced target-mediated drug disposition (TMDD), FLT3L, first-in-human (FIH) dose, model-informed drug development (MIDD)

## Abstract

FLT3L-Fc is a half-life extended, effectorless Fc-fusion of the native human FLT3-ligand. In cynomolgus monkeys, treatment with FLT3L-Fc leads to a complex pharmacokinetic/pharmacodynamic (PK/PD) relationship, with observed nonlinear PK and expansion of different immune cell types across different dose levels. A minimal physiologically based PK/PD model with expansion-enhanced target-mediated drug disposition (TMDD) was developed to integrate the molecule’s mechanism of action, as well as the complex preclinical and clinical PK/PD data, to support the preclinical-to-clinical translation of FLT3L-Fc. In addition to the preclinical PK data of FLT3L-Fc in cynomolgus monkeys, clinical PK and PD data from other FLT3-agonist molecules (GS-3583 and CDX-301) were used to inform the model and project the expansion profiles of conventional DC1s (cDC1s) and total DCs in peripheral blood. This work constitutes an essential part of our model-informed drug development (MIDD) strategy for clinical development of FLT3L-Fc by projecting PK/PD in healthy volunteers, determining the first-in-human (FIH) dose, and informing the efficacious dose in clinical settings. Model-generated results were incorporated in regulatory filings to support the rationale for the FIH dose selection.

## 1. Introduction

Cancer immunotherapy (CIT) boosts the immune system’s ability to identify and eliminate cancer cells. Checkpoint inhibitors including anti-programmed death-1 (PD-1) or anti-programmed death-ligand 1 (PD-L1) therapies such as atezolizumab, nivolumab, and pembrolizumab, have been approved for several oncology indications. However, the majority of patients do not respond to treatment with checkpoint inhibitors, or they only experience transient disease stabilization. Thus, a significant unmet need persists for patients with metastatic solid tumors [1,2,3] necessitating the application of innovative and informed strategies such as model-informed drug development (MIDD) for effective dose-finding and the efficacy prediction of novel molecules. It has been hypothesized that low dendritic cell (DC) counts within the tumor microenvironment could hinder the effectiveness of CIT therapy in cancer patients. Among DC subsets, conventional DC1s (cDC1s) reportedly have high intrinsic capacity to cross-present internalized antigens via MHC class I, enabling CD8^+^ T cell activation and inducing anti-tumor T cell-mediated immunity [4,5,6,7]. cDC1s drive the initiation of the cancer immunity cycle by priming T cells and maintaining antigen-specific T cell recall responses [8,9,10]. cDC1s are scarce within tumors, due in part to the immunosuppressive nature of the tumor microenvironment which leads to suboptimal recruitment, differentiation, or viability of cDC1s [11,12]. Conversely, an abundance of cDC1s in the tumor microenvironment has been associated with increased T cell infiltration, the response to checkpoint inhibitors, and overall survival rates in patients with cancer [13,14,15].

Fms-related tyrosine kinase 3 ligand (FLT3L) is a soluble growth factor (or cytokine) that plays a critical role in regulating hematopoiesis. FLT3L functions by agonizing the fms-related tyrosine kinase 3 receptor (FLT3; also known as CD135), which is expressed on DCs as well as myeloid and lymphoid progenitor cells [16,17,18,19,20]. Agonism of this pathway initiates signaling cascades that promote DC proliferation, differentiation, mobilization, and survival [21,22,23,24,25,26,27]. FLT3L treatment has been shown to expand not only cDC1s, but also conventional type 2 DCs (cDC2s), plasmacytoid DCs (pDCs), and other immune cells of myeloid origin in blood, lymphoid organs, and tumors [28,29,30,31]. Preclinical proof-of-concept studies have demonstrated that FLT3L treatment improves anti-tumor efficacy by enhancing T cell priming [32,33,34,35,36,37,38], providing a rationale for combining FLT3L therapy with T cell-directed CIT including checkpoint inhibition.

Clinical studies have demonstrated that targeting the FLT3 signaling cascade is a promising approach to safely expanding DCs and enhancing anti-cancer immunity in patients. CDX-301, a recombinant human FLT3L molecule (Celldex Therapeutics, Inc., Hampton, NJ, USA), was tested as a single agent in healthy volunteers, as well as in combination with immune checkpoint inhibitors, radiation therapy, and other innate immune-activating agents in phase I/II clinical trials [23,34,36,39]. In healthy volunteers, CDX-301 was well-tolerated at 25 μg/kg/day for 7 or 10 days and 75 μg/kg/day for 5 days [23]. In another study, recombinant human FLT3L was well-tolerated when dosed up to 100 μg/kg/day for 14 days [22]. GS-3583, another FLT3L agonist with a crystallizable fragment (Fc) fusion protein molecule (FLT3L-Fc) (Gilead Sciences, Inc., Foster City, CA, USA), was recently tested in healthy volunteers as a single dose and found to be well-tolerated up to 2000 μg (40). Both recombinant and Fc fusion FLT3L formats demonstrated significant expansion of DCs in healthy volunteers [22,23,40] and, in combination with adjuvants, can potentially promote T cell-mediated anti-tumor activity in cancer patients. The PK properties of recombinant human FLT3L molecules, e.g., CDX-301, necessitate daily dosing, presenting logistical considerations that potentially impede the broader application of FLT3L-targeting treatments in clinical settings.

FLT3L-Fc (Genentech, Inc., South San Francisco, CA, USA) is a fusion of the truncated extracellular domain of human FLT3L and the Fc portion of human IgG1. The Fc portion has been further engineered to impair binding to human Fcγ receptors, thereby attenuating Fc-mediated effector functions [41]. The FLT3L-Fc molecule has been designed to extend the FLT3L half-life and decrease the clinical dosing frequency required to achieve therapeutically significant cDC1 expansion. It thus has the potential to enhance anti-tumor immunity, especially when used in combination with DC maturation agents and checkpoint inhibitor immunotherapy.

A minimal physiologically based pharmacokinetic/pharmacodynamic (PBPK/PD) model [42,43] with expansion-enhanced target-mediated drug disposition (TMDD) was developed to (1) integrate the mechanism of action of FLT3L-Fc, (2) explain and provide insight into the complex PK/PD relationship of FLT3L-Fc and CDX-301 observed in preclinical and clinical settings, and (3) provide a rationale for the selection of the first-in-human (FIH) dose in healthy volunteers. The model was first used to capture PK of FLT3L-Fc in cynomolgus monkeys and then used to project clinical PK by translating model parameter values, including clearance, physiological volumes, and flow rates [42,43,44], and accounting for clinical target-related parameters. Given that FLT3L-Fc and CDX-301 are expected to exhibit comparable PK/PD relationships due to their similar affinities to FLT3 receptors and mechanisms of action [22,23], the clinical PK/PD data from healthy volunteers treated with CDX-301 were incorporated into the model and used to project cDC1 or total DC expansion in peripheral blood after administration of FLT3L-Fc to healthy volunteers. This work describes the model and the MIDD strategy used to project a FIH dose aimed at achieving a minimum pharmacologically active dose (mPAD) level with an adequate safety margin for a Phase IA study (clinical trial number: 92655801, www.isrctn.com, accessed on 26 April 2024), in order to support clinical study design and regulatory submissions.

## 2. Materials and Methods

### 2.1. Ethics Statement

The authors confirm that they have obtained appropriate institutional review board approval for all animal experimental investigations at Charles River Laboratories (CRL, Reno, NV, USA and Sherbrooke, QC, Canada). All procedures were approved by the Institutional Animal Care and Use Committee (IACUC) at CRL (protocol code 20239581, 30 March 2020; protocol code 5003546, 7 December 2020) and were performed in compliance with the Animal Welfare Act, the Guide for the Care and Use of Laboratory Animals, and the Office of Laboratory Animal Welfare.

### 2.2. Datasets Used in the Workflow of Minimal PBPK/PD Model Development

Preclinical and clinical data were obtained from the published literature or the studies conducted by Genentech. A summary of the data used for model calibration or validation, including study titles, dose levels/regimens, measurements, and the source of the data is provided in Appendix A. The list of studies includes two internal preclinical studies in cynomolgus monkeys using FLT3L-Fc (Genentech Studies) and three clinical studies from other FLT3L-based therapies [22,23,40]. The PK of FLT3L-Fc in preclinical studies 1 and 2 (Genentech Studies) were measured using ELISA methods (anti-FLT3L capture and anti-human Fc detection) with a lower limit of quantification (LLOQ) of 31.3 ng/mL for preclinical study 1 and 7.8 ng/mL for preclinical study 2. Detection of anti-drug antibodies (ADAs) to FLT3L-Fc for preclinical study 1 utilized a direct ELISA where FLT3L-Fc was directly coated on the ELISA plate, and cynomolgus monkey IgG directed against FLT3L-Fc was detected with an anti-cynomolgus monkey IgG antibody. Preclinical study 2 utilized a bridging ELISA with biotin- and digoxigenin-conjugated RO7497987 to capture and detect ADAs directed against RO7497987. The PK/PD of CDX-301 and recombinant human FLT3L in clinical studies 1 and 2, and the PK of GS-3583 in clinical study 3 were digitized using “GetData Graph Digitizer” version 2.26 (http://getdata-graph-digitizer.com, accessed on 26 April 2024). Since other recombinant human FLT3Ls are expected to behave similarly to CDX-301, we will henceforth refer to them collectively as CDX-301 for the purpose of our modeling. Figure 1 (created with Biorender) provides a summary of the workflow and the use of data for model calibration and validation. 

### 2.3. Model Structure and Key Assumptions

#### 2.3.1. PK Model Structure of FLT3L-Fc and GS-3583

A minimal PBPK model with expansion-enhanced TMDD was developed to describe the available FLT3L-Fc PK data from cynomolgus monkeys and project FLT3L-Fc PK profiles (Figure 1A). The minimal PBPK model structure and modeling assumptions were previously described by Jusko et al. [42,45]. Briefly, the minimal PBPK model integrates physiological factors and divides tissue spaces into two groups based on their vascular endothelial structure, leaky and tight, which subsequently provides two general sites of distribution. The model also includes convection as the primary mechanism of antibody trafficking from plasma into tissues and return to plasma [42,45]. In this work, free FLT3L-Fc or GS-3583 in the central compartment binds to free FLT3 receptors, forming single-bound or double-bound complexes with the receptors. Subsequently, homodimer (double-bound) complexes induce signaling cascades that promote the expansion of the available target. The differential equations describing the minimal PBPK model with expansion-enhanced TMDD are shown below:(1)$$\frac{dC_{p}}{dt}=\frac{Dose}{V_{p}}-{kon}_{1}\cdot C_{p}\cdot C_{R}+{koff}_{1}\cdot C_{Rp}\;+(C_{lymph}\cdot L-C_{p}\cdot L_{tight}\cdot \left(1-\sigma_{tight}\right)\;-C_{p}\cdot L_{leaky}\cdot \left(1-\sigma_{leaky}\right)-C_{p}\cdot {CL}_{p})/V_{p}$$(2)$$\frac{dC_{R}}{dt}=ksyn-kdeg\cdot C_{R}-{kon}_{1}\cdot C_{p}\cdot C_{R}+{koff}_{1}\cdot C_{Rp}-{kon}_{2}\cdot C_{R}\cdot C_{Rp}\;+{koff}_{2}\cdot C_{RpR}+C_{R_{TOT}}\cdot vm_{prolif}\cdot \left[\frac{{\left(\frac{RO_{DB}}{100}\right)}^{\alpha }}{{\left(\frac{RO_{DB}}{100}\right)}^{\alpha }+km_{prolif}^{\alpha }}\right]$$(3)$$\frac{dC_{Rp}}{dt}={kon}_{1}\cdot C_{p}\cdot C_{R}-{koff}_{1}\cdot C_{Rp}-{kon}_{2}\cdot C_{R}\cdot C_{Rp}\;+{koff}_{2}\cdot C_{RpR}-kdeg\cdot C_{Rp}$$
(4)$$\frac{dC_{RpR}}{dt}={kon}_{2}\cdot C_{p}\cdot C_{Rp}-{koff}_{2}\cdot C_{RpR}-kdeg\cdot C_{RpR}$$
where *C_p_* is the drug concentration in the central compartment, and *C_tight_*, *C_leaky_*, and *C_lymph_* are drug concentrations in the tight, leaky, and lymph compartments, respectively. *V_p_* denotes the species-specific volume of the central compartment, *V_tight_* represents the interstitial fluid volumes of tissues with non-fenestrated capillaries (e.g., muscle, skin, adipose, and brain), and *V_leaky_* denotes interstitial fluid volumes of tissues with fenestrated capillaries (e.g., heart, liver, kidney, etc.) [46]. *L_tight_* and *L_leaky_* are the flows to collective tight and leaky tissues, and L represents the total lymph flow equal to the sum of *L_tight_* and *L_leaky_*. Parameters *σ_tight_* and *σ_leaky_* are vascular reflection coefficients for tight and leaky tissues, and *σ_lymph_* is the lymphatic reflection coefficient [42]. *CL_p_* denotes the non-specific clearance from the plasma. *C_R_*, *C_Rp_*, and *C_RpR_* represent concentrations of the free, single-bound, and double-bound forms of the FLT3 target. *C_R_* is assumed to be at homeostasis in the absence of the drug and is synthesized and degraded at zero-order and first-order rate constants *ksyn* and *kdeg*, respectively. *C_p_* binds to *C_R_* and dissociates from *C_Rp_* via rate constants *kon*_1_ and *koff*_1_, respectively. *C_Rp_* binds to *C_R_* and dissociates from *C_RpR_* via rate constants *kon*_2_ and *koff*_2_, respectively. Note that it is assumed that the degradation rate constant for the complexes is the same as the degradation rate constant for the free receptor. Complex formation between FLT3L-Fc and two FLT3 receptors leads to downstream intracellular signaling and subsequent expansion of the total pool of FLT3 receptors [23,47]. In the model, the percentage of FLT3 receptors bound in a double-bound complex (*RO_DB_* or double-bound receptor occupancy) expands the pool of free FLT3 receptors. Parameter *vm_prolif_* represents the maximum rate of FLT3 receptor expansion, *km_prolif_* denotes the percentage of FLT3 *RO_DB_* to achieve 50% of maximum expansion, and *α* represents the Hill coefficient. *RO_DB_* is calculated as follows: (5)$$RO_{DB}=\left(\frac{2\cdot C_{RpR}}{C_{R_{TOT}}}\right)\times 100$$
(6)$$C_{R_{TOT}}=C_{R}+C_{Rp}+2\cdot C_{RpR}$$
where *C_RTOT_* represents the total pool of FLT3 receptors. *RO_SB_* and *RO* represent the percentage of FLT3 receptors bound in a single-bound complex and the percentage of the total FLT3 receptors bound, respectively, and are calculated as follows:(7)$$RO_{SB}=\left(\frac{C_{Rp}}{C_{R_{TOT}}}\right)\times 100$$
(8)$$RO=\left(1-\frac{C_{R}}{C_{R_{TOT}}}\right)\times 100$$

#### 2.3.2. PK Model Structure of CDX-301

A standard single-compartment PK model with subcutaneous absorption and nonlinear elimination was used to describe the measured concentration of CDX-301 from healthy volunteers (Figure 1B). The CDX-301 PK model is shown below:(9)$$\frac{dA_{SC,CDX}}{dt}={Dose}_{CDX}-kabs\cdot A_{SC,\;CDX}$$
(10)$$\frac{dA_{C,\;CDX}}{dt}=kabs\cdot A_{SC,CDX}-\frac{V_{m,\;CDX}\cdot C_{C,CDX}}{K_{m,\;CDX}+C_{C,CDX}}$$
(11)$$C_{C,CDX}=\frac{A_{C,CDX}}{V_{d,CDX}}$$
where *A_SC,CDX_* and *A_C,CDX_* represent the amount of CDX-301 in the subcutaneous and central compartments, respectively, and *C_C,CDX_* represents the concentration of CDX-301 in the central compartment. *k_abs_* denotes the absorption rate of CDX-301 from the subcutaneous compartment into the central compartment. *V_d,CDX_* denotes the volume of the central compartment. *V_m,CDX_* represents the maximum nonlinear elimination rate of CDX-301, and *K_m,CDX_* denotes the concentration of CDX-301 needed to achieve 50% of *V_m,CDX_*.

#### 2.3.3. PD Model Structure for DC Expansion

FLT3L-Fc and CDX-301 are assumed to elicit the same exposure response on cDC1 and cDC2 expansion [22,23], and, hence, concentrations of FLT3L-Fc and CDX-301 in nM can be used interchangeably to drive cell expansion (Figure 1C). The expansion of immune cells increases the total FLT3 receptors, thereby enhancing the TMDD. The expansion-enhanced TMDD was previously modeled using an approach that decouples PK and PD [48,49]. The current model adopts this decoupled approach wherein unbound drug concentration drives a delayed expansion of cDCs following the initiation of dosing via a series of transit compartments. *cDC_x_* (*x* takes values of 1 and 2 to represent cDC1 or cDC2, respectively) proliferates in a drug-dependent and target-dependent manner [23], as represented by the following equation: (12)$$\frac{d{cDC}_{x}}{dt}=kdeg_{x}\cdot {cDC}_{x,0}-kdeg_{x}\cdot {cDC}_{x}\;+{cDC}_{x}\cdot kdeg_{x}\cdot \frac{{vm}_{x}\cdot {\left(C3_{x}\right)}^{n_{x}}}{{\left({km}_{x}\right)}^{n_{x}}+{\left(C3_{x}\right)}^{n_{x}}}\;-f_{DC1}\cdot kdeg2\cdot \frac{{max\left({cDC}_{x}-{cDC}_{x,0},0\right)}^{2}}{{cDC}_{x,0}}$$
where *cDC_x,_*_0_ represents *cDCx* counts at baseline and *C*3*_x_* represents the drug-concentration in the third transit compartment to induce expansion of *cDC_x_* cells. *vm_x_* represents the maximum rate of drug-induced expansion of *cDC_x_* cells, *km_x_* denotes the drug concentration required to achieve 50% of *vm_x_*, and *n_x_* represents the Hill coefficient. *kdeg*2 is a second-order rate constant for apoptosis of *cDCx* cells. The term *f_DC_*_1_ only appears in the equation for *cDC1* and denotes the fractional term that relates the second-order rate constant for apoptosis *cDC1* to that of *cDC2*. In the figures, we show *cDC1* and total *DC*, which is the sum of *cDC1* and *cDC2*.

### 2.4. Overview of Model Calibration, Validation, and Simulation

As previously described, a mPBPK/PD model with expansion-enhanced TMDD in the central compartment was used to capture FLT3L-Fc preclinical data and project clinical PK/PD in healthy volunteers. The model-building strategy consists of eight steps (Figure 1D): The minPBPK model was calibrated to FLT3L-Fc PK data from cynomolgus monkeys (0.1 to 10 mg/kg; preclinical study 1).Simulations using the fitted model parameters were validated against a second cynomolgus monkey PK dataset (1 and 3 mg/kg; preclinical study 2).Cynomolgus monkey PK parameters were translated to human PK parameters using the appropriate human non-specific clearance, physiological volumes, and flows to provide clinical PK predictions based solely on interspecies translation (the cyno-derived scenario).In parallel, a separate nonlinear single-compartment PK model was built to capture FLT3L (CDX-301) clinical PK data across the range of doses evaluated in healthy volunteers (3 μg/kg to 75 μg/kg once daily subcutaneous (SC) injections for 5, 7 or 10 consecutive days; clinical study 1).Using the CDX-301 PK model, parameters of human PD were calibrated to the published time-course data on cDC1 (clinical study 1) [23] and total DC counts (clinical study 2, part 1) [22].Human PD simulations were validated using total DC expansion data in healthy volunteers following a range of FLT3L doses (10 μg/kg to 100 μg/kg SC injections once daily for 14 consecutive days; clinical study 2, part 2) [22].Alternative clinical PK predictions of FLT3L-Fc were projected using clinical PK data from subjects treated with a single-dose FLT3 agonist Fc fusion protein molecule, GS-3583 (225 μg and 675 μg; clinical study 3) [40] (human-derived scenario).Projected clinical PK from both cynomolgus monkey- and human-derived scenarios were used to predict cDC1 and total DC expansion for different doses of FLT3L-Fc.

The model was developed using the Simbiology^®^ toolbox in MATLAB^®^ (version R2018a, Mathworks Inc., Natick, MA, USA), and model parameterization/simulations were performed using the gQSPSim toolbox [50]. Particle swarm methodology [48] was used to calibrate model parameters to capture preclinical and clinical PK or PD measurements. The list of model parameters, rate constants, and variables for preclinical and clinical simulations is provided in Appendix A.

### 2.5. Data Availability

The datasets generated and/or analyzed during the current study are available from the corresponding author upon reasonable request.

### 2.6. Code Availability

The code for execution of the model, along with the corresponding ordinary differential equations (ODEs), are available in the Appendix A.

## 3. Results

### 3.1. Brief Description of the PK/PD Model of the FLT3L-Based Treatments

The minimal PBPK/PD model with expansion-enhanced TMDD was designed to capture FLT3L-Fc PK data in cynomolgus monkeys and Gilead’s GS-3583 PK data in healthy volunteers, including the effect of the drug-induced expansion of the target. The binding of FLT3L-Fc to one or two monomeric FLT3 receptors in circulation accounted for the observed TMDD. Homodimer binding drives the proliferation of immune cells, which leads to the expansion of the total FLT3 receptors on DCs and other bone marrow common progenitors, all of which were lumped into one total target capacity. The PK of CDX-301, which is administered as daily SC injections, was captured empirically using a single-compartment model with nonlinear clearance; the simple PK model was used for CDX-301 because its short half-life hinders the estimation and identification of TMDD-related parameters and non-specific clearance. After capturing the PK for both modalities, the CDX-301 PK was linked to an indirect PD response model via a series of transit compartments to capture the observed expansion of cDC1s and total DC cells. Exposure to the drug results in the first-order proliferation of the cells, whereas both first- and second-order apoptosis terms are used to capture the observed plateau in PD and the subsequent rapid decay once the drug clears. The model structure is depicted in Figure 1.

### 3.2. Calibration and Validation in Nonclinical Species to Establish Confidence in the Model

Surface Plasmon Resonance (SPR) binding analysis of FLT3L-Fc demonstrated comparable affinities to the recombinant human and cynomolgus monkey FLT3 receptors, which supports cynomolgus monkeys as a relevant test species for evaluation of FLT3L-Fc PK and its translation to humans. PK data in cynomolgus monkeys and the corresponding model fits are provided in Figure 2A. Plasma concentrations of FLT3L-Fc exhibited nonlinear PK within the tested dose range of 0.1 to 10 mg/kg. Anti-drug antibodies (ADAs) to FLT3L-Fc were detected by day 14 in all nine animals in preclinical study 1 (Figure 2A), and in all 12 animals in preclinical study 2 (Figure 2B). The dose-normalized area under the curve from days 0 to 21 (AUC_0–21_) for 0.1 to 10 mg/kg ranged from 7 ± 0.497 to 1660 ± 336 µg/mL·day, and corresponding total clearance (CL) ranged from 13.7 ± 0.901 to 4.92 ± 1.96 mL/day/kg. The dose-normalized AUC_0–21_ and CL were observed to be dose-dependent, suggesting nonlinear PK [50]. As shown in Figure 2A, for 10 mg/kg, the model successfully captured the C_max_ and the linear slow elimination phase for ADA-negative PK samples but did not capture ADA-positive samples, particularly near the end of the study, as expected. The model also captured the nonlinear portion of the PK (for ADA-negative time points) well for lower doses at 0.1 and 1 mg/kg. The predictive capability of the model was verified by comparing simulations at 1 and 3 mg/kg doses with PK data from preclinical study 2 (Figure 2B). Measured concentrations of ADA-positive samples were again lower than predicted at later time points, likely reflecting the impact of ADA on exposure. However, the model was able to capture serum plasma levels in ADA-negative samples, which provided confidence in the predictive capabilities of the PK model for nonclinical species. The model was then used to simulate the dynamics of FLT3 receptor concentrations and receptor occupancy following FLT3L-Fc treatment in cynomolgus monkeys (Figure 3). Simulations suggest that increasing the dose from 0.01 to 10 mg/kg leads to increased total occupancy of FLT3 receptors in single-bound or double-bound complexes. At 1 mg/kg and higher, the model predicts near-complete receptor occupancy. Unlike single-bound receptor occupancy, which exhibited a dose-dependent increase, the percentage of double-bound receptor occupancy reached its peak at 0.1 mg/kg (83%) and declined following doses ranging from 1 to 10 mg/kg. This reflects a shift from double-bound complexes towards single-bound ones at higher drug levels.

### 3.3. The Model Reproduces Clinical PK and PD from Healthy Volunteers Treated with FLT3L (CDX-301)

A single-compartment PK model with nonlinear clearance was used to capture the observed PK of CDX-301 in healthy volunteers treated with daily SC injections at 3, 10, 25, and 75 μg/kg/day in clinical study 1 [23] (Figure 4A). CDX-301 PK exhibited a dose-linear increase in the maximum observed concentration (C_max_) and a dose-dependent increase in the area under the curve (AUC) [23]. The PD model parameters were fitted to the cDC1 time-course data from Anandasabapathy et al. (clinical study 1) [23] and total DC counts (equivalent to sum of cDC1s and cDC2s in the model) from Maraskovsky et al. (clinical study 2, part 1) [22]. A subset of total DC data from Maraskovsky et al. [22] (clinical study 2, part 2) was set aside for validation. The model simulations successfully reproduced clinical CDX-301 PK and captured the dynamics and dose-dependent saturable expansion of cDC1 and total DC counts reasonably well across both studies (Figure 4). Given that FLT3L-based therapies lead to a significant fold expansion of DCs, the group-specific baseline levels for cDC1 were incorporated into the model to improve the goodness of fit. In clinical study 1, CDX-301 daily injections from 3 to 75 μg/kg produced a dose-dependent expansion of cDC1s (Figure 4B). The model captured the dynamics of expansion for all groups. Although the model slightly overpredicted the peak of cDC1 expansion in the 25 μg/kg × 5 cohort and underpredicted the peak in the 25 μg/kg × 10 cohort, it successfully captured peak expansion for the upper (75 μg/kg × 5) and lower dose regimens (3 and 10 μg/kg × 5) (Figure 4B). Conversely, in clinical study 2, total DC expansion was comparable between the lowest dose (10 μg/kg/day) and the highest dose (100 μg/kg/day) following 14 days of daily treatments. Further, the total DC profile showed signs of saturation at around 10 days post initiation of FLT3L treatment (Figure 2D and Figure 4C). The model captured the rate of total DC expansion and the magnitude and timing of saturation following treatment with 100 μg/kg/day for 14 days (Figure 4C) and successfully predicted the saturation from 10–100 μg/kg (Figure 4D). These calibration and validation steps established confidence in the predictive capabilities for clinical PD projection.

### 3.4. The Model Projected Clinical PK and PD for FLT3L-Fc Treatment across Dose Levels

Cynomolgus monkey PK of FLT3L-Fc was used to project human PK by translating cyno-derived physiologically based PK parameters including lymph flow rate, interstitial fluid volume, and volume of distribution to human parameters and allometrically scaling CL, as previously published [42,43]. Two PK scenarios were simulated using different target-related parameters (i.e., FLT3 target capacity and proliferation-related parameters): (1) in the cyno-derived scenario, parameters were fixed based on the estimated parameters from cynomolgus monkey PK data or (2) in the human-derived scenario, parameters were adjusted to better match the PK of limited PK data of GS-3583 in healthy volunteers at two dose levels (Figure 5). The model simulated plasma exposures following multiple doses at levels ranging from 0.01–1 mg/kg for both scenarios. These PK profiles were then used to project the dynamics and level of cDC1 and total DC expansion using the PK/PD relationship established from the CDX-301 clinical data. Nonlinear PK was apparent in both scenarios at ≤0.1 mg/kg, and the loss of exposure due to TMDD at low dose levels was more pronounced in the cyno-derived scenario than the human-derived scenario because of the greater target concentration in the cyno-derived scenario (Figure 6A,D). As a result, the cyno-derived scenario required about a three-fold greater dose than the human-derived scenario to maximize the DC expansion effect (0.3 mg/kg vs. 0.1 mg/kg), causing cDC1 and total DC to plateau at approximately 70,000 and 1,000,000 cells/mL, respectively (Figure 6). Given that the human-derived scenario required a lower dose to induce a similar PD effect than the cyno-derived scenario, the human-derived scenario was chosen for FIH dose selection as the more conservative approach to achieve mPAD in healthy volunteers. 

### 3.5. Rationale for FIH Dose Selection in Healthy Volunteers

A model-based FIH dose for FLT3L-Fc was selected based on an mPAD approach and supported with safety margins determined from FLT3L-Fc preclinical toxicity data. The criteria for the FIH dose were set to result in less than 20% of maximal potential peripheral blood cDC1 expansion. Assuming that the typical patient weighs 70 kg, a starting dose of 700 μg (0.01 mg/kg) was estimated to induce a peak expansion for peripheral blood cDC1 at 9500 cells/mL (about 8-fold expansion), which is approximately 13% of the maximum expansion level for cDC1 (Figure 6E). For the proposed FIH dose, the maximum total RO in the central compartment is estimated at 67%, with an average RO over 21 days of about 16% (Appendix A). In addition, the projected clinical exposure at the FIH dose level was estimated to be within the range of measured exposures following CDX-301 administration to healthy volunteers [23]. Furthermore, the exposures at the 3 mg/kg no-observed-adverse-effect level (NOAEL) from the cynomolgus monkey GLP-compliant toxicity study provided safety margins of more than 220-fold and more than 1500-fold over the clinical C_max_ and AUC_0–21_, respectively, at the FIH dose [50]. The selected FIH dose also enables a full characterization of the PK/PD of FLT3L-Fc in healthy volunteers across a reasonable number of cohorts.

The model-based FIH dose selection and justification were communicated with the FDA in the regulatory submission, and the molecule is currently being tested in healthy volunteers (clinical trial number: 92655801, www.isrctn.com, accessed on 26 April 2024). The clinical simulations and adequate preclinical safety margins collectively supported the mPAD-based starting dose of 700 μg FLT3L-Fc IV for phase Ia clinical trials.

## 4. Discussion

Therapeutics that expand target-expressing cell populations are likely to exhibit expansion-enhanced TMDD, whereby higher doses drive greater expansion and thus increased total drug clearance [48,49]. The mechanistic minimal PBPK/PD model presented here integrates complex PK/PD relationships observed in preclinical and clinical studies of multiple FLT3L-based therapies. This model enabled the projection of clinical PK/PD profiles for FLT3L-Fc to inform clinical trial design, and was effectively utilized in our regulatory documents to propose the FIH dose selection for the clinical trial (clinical trial number: 92655801, www.isrctn.com, accessed on 26 April 2024). PK/PD was simulated based on the relationship observed from an extensive range of FLT3L (CDX-301) doses (3 μg/kg for 5 days to 100 μg/kg for 14 days) (Figure 4), encompassing a range of projected clinical exposures of FLT3L-Fc following FIH dosing. These clinical simulations provided quantitative guidance for the selection of 700 μg (0.01 mg/kg) as a safe starting dose in a phase Ia clinical trial (supported by adequate safety margins), while inducing an approximately 8-fold expansion of cDC1s in human subjects and avoiding exposure of human subjects to sub-pharmacological dose levels (clinical trial number: 92655801, www.isrctn.com, accessed on 26 April 2024). This approach and model-informed FIH dose selection were communicated with the FDA in the Investigator’s Brochure (IB) and the phase 1 protocol.

Nonlinear TMDD-driven PK behavior for FLT3L-Fc is expected in the phase Ia clinical trial based on the nonlinear PK observed in cynomolgus monkeys and the PK nonlinearity will likely be influenced by the level of target expansion observed in healthy volunteers. Cynomolgus monkeys are an appropriate animal model for clinical translation because human non-specific clearance can generally be well-predicted from observed PK in this species [44]. However, PK simulations translated from cynomolgus monkeys underpredicted GS-3583 PK data in healthy volunteers (Figure 5, dashed lines), suggesting that the target expression and other target-related parameters between humans and cynomolgus monkeys may not be equivalent. Reducing FLT3 receptor concentrations by 2.8-fold and adjusting the proliferation dynamics of FLT3 improved the human GS-3583 PK predictions, better capturing the initial and terminal time points in healthy volunteers (Appendix A, Figure 5). In addition, these revised simulations led to a higher receptor occupancy and cDC1 and total DC expansion for a given FLT3L-Fc dose (Appendix A), providing a more conservative mPAD-based starting dose (0.01 mg/kg) compared to a starting dose selected based on the cyno-derived scenario (0.03 mg/kg).

The FIH starting dose of 700 μg (0.01 mg/kg) was selected to minimize the risk of potential adverse events in trial participants while ensuring that healthy volunteers are not subjected to sub-pharmacological exposures. The potential adverse effects of FLT3L-Fc are informed by findings from preclinical toxicity studies and the clinical safety profiles of previously studied FLT3L-based therapies. FLT3L-Fc was well-tolerated in cynomolgus monkeys up to 10 mg/kg q3w (with a NOAEL of 3 mg/kg identified in the GLP-compliant toxicity study), and the findings were generally consistent with the pharmacodynamic profile of FLT3L-Fc [50]. The clinical FLT3L-based molecules, CDX-301 and GS-3583, were also well-tolerated in healthy volunteers at comparable exposures to the proposed FIH starting dose [23,40]. A fraction of healthy volunteers (29%) receiving CDX-301 at a dose ≥ 25 μg/kg exhibited transient lymphadenopathy [23], suggesting the possibility of infiltration and expansion of histiocytes and leukocytes in lymphoid tissues. In FLT3L-Fc clinical studies, all study participants will be monitored for lymphadenopathy, splenomegaly, and organ enlargement. GS-3583 was well-tolerated as a single dose in healthy volunteers up to 2000 μg (or about 0.029 mg/kg assuming a 70 kg patient) with no reported serious grade 3 adverse effects [40]. The FIH dose of FLT3L-Fc is predicted to produce 8-fold cDC1 expansion, which is less than 20% of the maximum expansion observed in subjects treated with CDX-301. Furthermore, simulations suggest that DC counts are expected to return to baseline within the trial duration (Appendix A), minimizing the risk of extended immune cell expansion in trial participants. Collectively, preclinical data with FLT3L-Fc from the repeat-dose toxicity studies, available clinical data from FLT3L-based therapies, and model simulations support the proposed FIH dose for entry into the clinic.

The nonlinear PK observed in cynomolgus monkeys was reproduced using a minimal PBPK model that captured expansion-enhanced TMDD, but the model did not account for the impact of ADA on exposure due to lack of preclinical-to-clinical translation. ADAs were detected in all animals on or after day 14 and generally impacted FLT3L-Fc exposures in cynomolgus monkeys with high titers. However, the ADA response to biological therapeutics in nonclinical species has a generally low predictive power for estimating immunogenicity in humans [51,52,53]. Furthermore, no ADAs were detected on day 21 in healthy volunteers receiving daily treatments of CDX-301 [23]. Considering FLT3L-Fc is a fully human recombinant FLT3L directly fused to a native human Fc, it is not anticipated to result in significant immunogenicity in the clinical study.

Interspecies PK translation of non-specific clearance for large molecules is relatively well-studied in predicting exposure in humans [42,43,44], but, because the system-specific factors can vary between species, translating TMDD, let alone expansion-enhanced TMDD, and predicting PD response are more challenging and not as robust [54]. Interspecies differences in PD were apparent in the magnitude of response and expansion profiles of cDC1 and total DCs, whereas mice, cynomolgus monkeys, and humans treated with FLT3L-based therapies exhibited substantially different PK/PD relationships. In mice, a dose-dependent expansion of cDC1s and total DCs was observed in animals treated with a mouse surrogate FLT3L-Fc at doses of 0.1 to 30 mg/kg with no signs of PD saturation or bell-shaped response (see Figure 3 by Decalf et al. [41]). In cynomolgus monkeys treated with 0.1, 1, and 10 mg/kg of FLT3L-Fc, observations show characteristics of PD saturation and a bell-shaped response (see Figure 4 in Wu et al., 2024 [50]), presumably because higher doses favor formation of binary (single-bound) complexes rather than the active ternary (double-bound) complexes. However, the collective clinical PD data in healthy volunteers treated with FLT3L-based therapies suggests a dose-dependent saturable PD response with no apparent bell-shaped behavior. Due to the uncertainty in the translatability of PK/PD between species, we primarily relied on the PK/PD relationship observed in humans treated with CDX-301 and projected clinical PK profiles under two scenarios: (1) directly translating PK from cynomolgus monkeys treated with FLT3L-Fc using cyno-based target-related parameters, and (2) translating PK-related parameters from cynomolgus monkeys in addition to incorporating clinical PK data from GS-3583 to estimate target-related parameters. The projected PK profiles were then used to predict cDC1 and total DC expansion under the two scenarios. In this way, PK/PD projections for healthy volunteers integrated preclinical and clinical PK data from FLT3L-Fc and GS-3583, respectively, with the clinical PK/PD relationship from CDX-301.

Like other engineered cytokines, the mechanism of action of FLT3L-Fc and the resulting PK/PD is complex. Upon binding of the molecule to two FLT3 receptors and forming a ternary complex, the complex induces several downstream pathways that lead to cell differentiation and proliferation in different tissues. The complex is also internalized, which leads to target-mediated drug disposition. Expansion of immune cells due to FLT3L-Fc treatment creates feedback that increases the total FLT3 target capacity and hence further augments TMDD for the drug. In this work, a decoupled approach [48,49] was used in model development to capture PK/PD data [48]. When employing the decoupled approach, it is crucial to utilize relevant PK/PD information, given that translation across species might be limited. Both coupled [49] and decoupled approaches have been used in modeling engineered cytokines. Coupling the PK and PD via FLT3 receptors could potentially provide more insight into mechanisms of drug activity. However, it requires extensive information on key system parameters (e.g., cell numbers, tissues of origin, receptor expression per cell type, and internalization rates of free and bound receptors, as well as the required threshold for the number of ligand-receptor complexes per cell type to induce the downstream pathways), complicating the model development by requiring elaborate, biological systems-based model structures used in quantitative system pharmacology (QSP) models. Further, the model predictions stand the risk of being inaccurate if the underlying data are not robust. Note that species differences across these parameters and their combined effect could lead to significant PK/PD differences and likely explains the complex PK and PD relationships observed in mice, cynomolgus monkeys, and humans. A decoupled approach is useful as it aims to capture PK and PD separately; in this modeling approach, an assumption is made that a total target capacity in the circulation can mimic the effects of FLT3 receptor expression on different cells in different tissues. Additionally, this total FLT3 target capacity can proliferate to mimic the effect of cell expansion in response to FLT3L-Fc treatment. Once the model captures the PK across different dose levels, the exposure can be used as the key driver of expansion for any cell type of interest, e.g., cDC1s and total DCs. This approach simplifies the biology, which involves the expansion of precursors in the bone marrow and differentiation to several immune cell populations. This model has several key practical advantages: the process of model calibration is more straightforward, and the model is more amenable to including other cell types without a significant need to re-calibrate the other aspects of the model. However, future QSP modeling could be utilized to properly account for key system parameters and improve the understanding of the role of FLT3L-based treatments in biological systems.

This work presents a model constructed using fixed physiological parameters with well-characterized values, effectively reducing concerns about parameter uncertainty. The key model parameters influencing PK and PD include target expression and PK parameters which were suitably fitted using data from cynomolgus monkeys. Validation was successfully performed using an independent dataset, which reinforced confidence in the generalizability and robustness to variations in parameter values. Additional model fitting and validation were performed using sparse and digitized clinical PD data from CDX-301. However, the model parameters can easily be refined to capture additional clinical data from FLT3L-Fc once they become available. FLT3L-Fc has the potential to improve and enhance anti-tumor immunity in combination with other treatments such as DC maturation agents and checkpoint inhibitors. Therefore, future uses of the model could include informing combination strategies to evaluate DC activation and anti-tumor efficacy in patients.

## Figures and Tables

**Figure 1 pharmaceutics-16-00660-f001:**
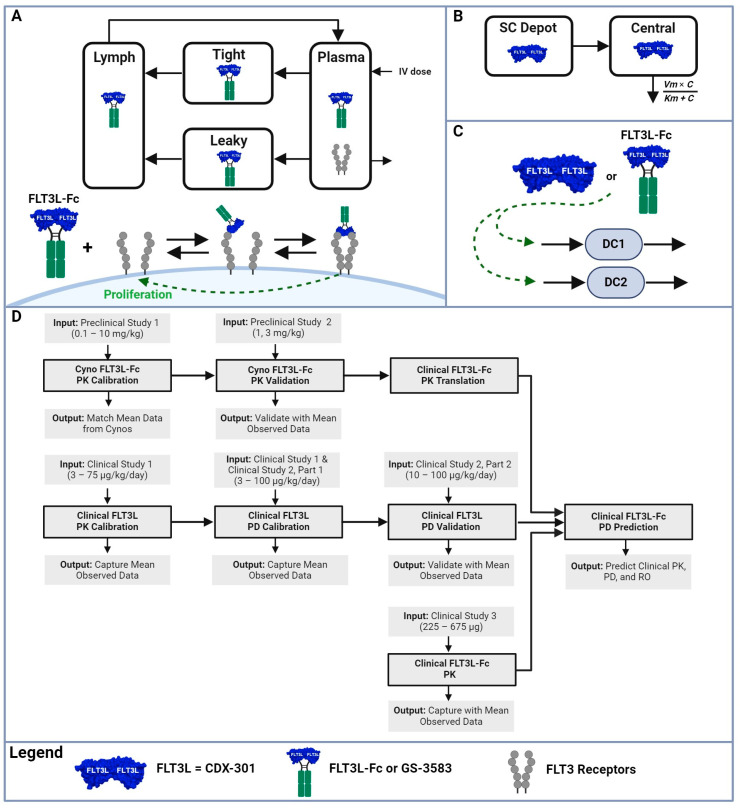
FLT3L-Fc minimal physiologically based PK/PD model and workflow. The minimal physiologically based pharmacokinetic (minPBPK) model and workflow describe the process of describing the in vivo dynamics of cDC1 and cDC2 following the interaction of FLT3L with the FLT3 receptors. (**A**) FLT3L-Fc and GS-3583 can bind to one or two FLT3 receptors in the central compartment to capture the impact of TMDD; the homodimer binding induces proliferation of different immune cells and leads to the expansion of the total target capacity of FLT3 receptors. (**B**) A one-compartment model with a subcutaneous compartment and a nonlinear clearance term describes the PK of FLT3L (CDX-301). (**C**) Free drug in the central compartment is linked to an indirect PD response model via a series of transit compartments to capture the observed expansion of cDC1 and cDC2 cells; dotted lines represent the indirect impact of the drug on cDC1 and cDC2 dynamics. (**D**) The model-building and dose projection workflow consist of eight steps described in the Section 2.

**Figure 2 pharmaceutics-16-00660-f002:**
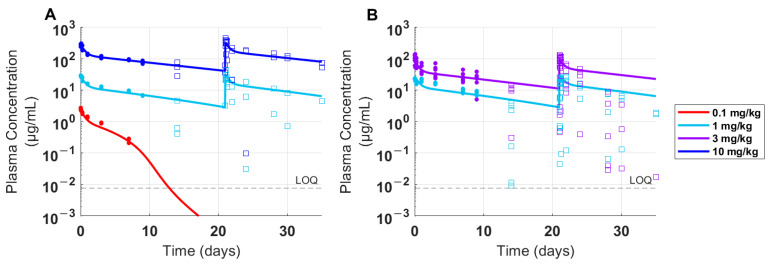
Time-course data and model fits for FLT3L-Fc PK in cynomolgus monkeys. Individual cynomolgus monkeys (*n* = 3) were treated with a single dose of 0.1 mg/kg, or two doses of 1 mg/kg, or 10 mg/kg with a three-week interval between doses in preclinical study 1. Individual cynomolgus monkeys (*n* = 6) were treated with two doses of 1 mg/kg, or 3 mg/kg with a three-week interval between doses in preclinical study 2. (**A**) Drug concentrations of FLT3L-Fc from preclinical study 1 was used to calibrate the cynomolgus monkey PK parameters in the model. (**B**) PK from preclinical study 2 served as a validation dataset for the model predictions. Open squares and solid dots represent individual PK samples that were ADA-positive and ADA-negative, respectively. LOQ: limit of quantification.

**Figure 3 pharmaceutics-16-00660-f003:**
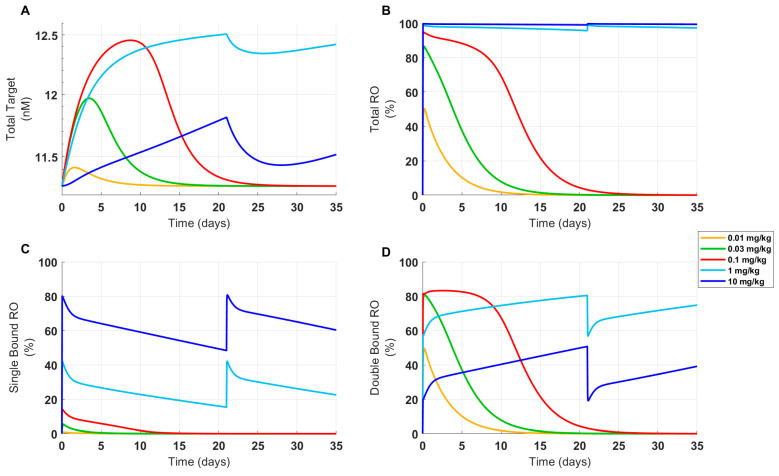
Projected target expression and receptor occupancy in cynomolgus monkeys following FLT3L-Fc treatment. (**A**) The total estimated target expression in cynomolgus monkeys following a single FLT3L-Fc dose (0.01, 0.03, and 0.1 mg/kg) or two FLT3L-Fc doses (1 and 10 mg/kg) administered three weeks apart is shown. (**B**) The percent of total FLT3 receptor occupancy (RO), (**C**) percentage of FLT3 receptors occupied in a single-bound complex, and (**D**) percentage of FLT3 receptors occupied in a double-bound complex are also shown.

**Figure 4 pharmaceutics-16-00660-f004:**
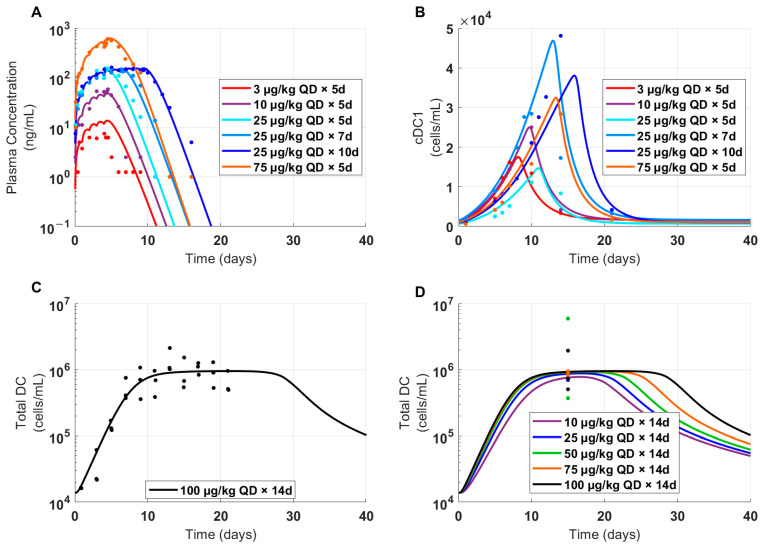
Clinical data and model outputs for PK, cDC1, and total DC counts in healthy human volunteers following treatment with FLT3L (CDX-301). PK/PD data from healthy human volunteers treated with FLT3L (CDX-301, Celldex Therapeutics, Inc., Hampton, NJ) was used to calibrate the model to capture the dynamics of cDC1 and total DC counts per mL following different FLT3L exposure levels. (**A**) Time-course CDX-301 concentration data were measured in subjects following the administration of daily SC injections of CDX-301. Data were digitized from clinical study 1 [23]. (**B**) Clinical PK of CDX-301 were linked to a PD model and parameters were fitted to digitized human cDC1 time-course data from clinical study 1 [23] and (**C**) total DC counts digitized from clinical study 2 [22]. (**D**) The model predictions were tested against total DC counts measured 15 days post FLT3L treatment from clinical study 2 (22). Dots show digitized data from clinical studies 1 and 2.

**Figure 5 pharmaceutics-16-00660-f005:**
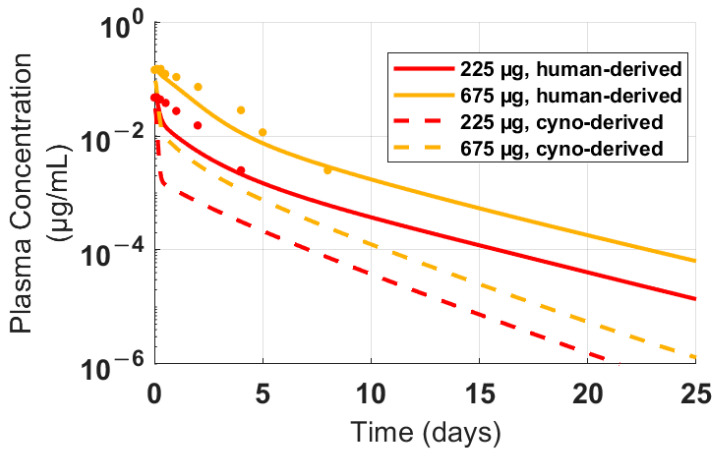
Clinical PK data and simulated PK profiles in healthy human volunteers following treatment with GS-3583. The FLT3L-Fc PK parameters were translated from parameters estimated using cynomolgus monkey PK data. FLT3 target-related parameters were estimated from cynomolgus monkeys (dashed line) or refined to improve the fit to human time-course data for the FLT3 agonist Fc fusion protein molecule (GS-3583, Gilead Sciences, Inc., Foster City, CA, solid line). The dots represent the mean concentration-time profiles of GS-3583 following a single IV administration of 225 and 675 μg to healthy human volunteers.

**Figure 6 pharmaceutics-16-00660-f006:**
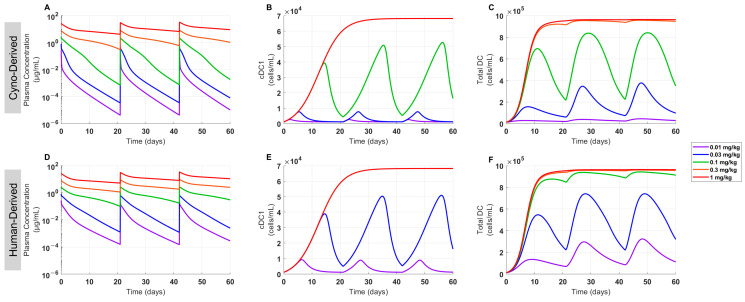
Projected time profiles of clinical PK, cDC1, and total DC cells in healthy volunteers treated with FLT3L-Fc. Simulations were performed to predict clinical FLT3L-Fc PK/PD profiles and compare cDC1 and total DC expansion under both cyno-derived and human-derived scenarios for a q3w dosing regimen. (**A**–**C**) In the first row, FLT3 target-related parameters were estimated from cynomolgus monkey PK, and (**D**–**F**) in the second row, FLT3 target-related parameters were estimated using human PK data from the FLT3 agonist Fc fusion protein molecule, GS-3583. Simulations were conducted for (**A**,**D**) PK, (**B**,**E**) cDC1 cells, and (**C**,**F**) total DC cells.

## Data Availability

Data is available upon request from the authors.

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
