# Peer review of "A Minimal PBPK/PD Model with Expansion-Enhanced Target-Mediated Drug Disposition to Support a First-in-Human Clinical Study Design for a FLT3L-Fc Molecule"

_pharmaceutics, 2024, doi:10.3390/pharmaceutics16050660_

Round 1

Reviewer 1 Report

Comments and Suggestions for Authors

Thank you for the opportunity to review this extremely interesting and successful manuscript.The authors describe the early clinical development of a promising substance (group), the FLT3 ligands. The development program presented is an example of how preclinical and early clinical data can be used in the sense of model-informed drug development (MIDD), in this case for dose selection in a clinical trial. The description is detailed and precise, the presentation clear and convincing, and the discussion balanced. Nevertheless, we have a few points that should be reconsidered.

Major comments

-          Introduction: The work is a precise description of the current problem (dose finding of a FLT3-Fc ligand). In addition, it is a good (and successful!) example of translating early information in the sense of a MIDD. Perhaps this aspect could be anchored even earlier (e.g. first paragraph) and even more prominently in the otherwise well-written introduction.

-          Methods: Please consider to already differentiate between coupled and uncoupled approaches within the methodology section (to introduce concepts later discussed in the discussion), e.g., by explicitly naming the chosen approach accordingly

-          Methods: The authors provide the workflow using Matlab Simbiology. This is helpful, especially for those with an active software license. However, to enable the adaptation in other software solutions (e.g. R/R-Studio with mrgsolve or rxode2) the model code is required. This cannot simply be called up (for us without a license, obviously). It therefore makes sense to also provide the model code in plain text so that it can be translated if necessary. Alternatively, the model reactions and ODEs could be provided in the supplement text (as is now the case in numerous other works created with Matlab Simbiology).

-          Discussion: please consider to emphasize right at the beginning that the presented work was successfully used for an actual dosing recommendation in an already started trial

-          Discussion: The authors describe the challenges of advanced models well (e.g., key system parameters). Please consider if a more mechanistic picture could be drawn that may include the uncertainties (mode of action, …) and yet unknown parameters / states / … as kind of a roadmap for the future of (QSP?) modeling of FLT3-based treatments.

Minor comments

-          Please consider numbering of the equations used

-          Figure 2: Legend appears to be cut in the printout

Reviewer 2 Report

Comments and Suggestions for Authors

This study was addressed to assess a pharmacokinetic/pharmacodynamic (PK/PD) model  for a FLT3L-Fc molecule (and other FLT3-agonist molecules, i.e.  GS-3583 and CDX-301) to project the expansion profiles of conventional DC1s and total dendritic cells (DC) in peripheral blood. Such study aimed to develop a strategy for clinical development of FLT3L-Fc in order to extend PK/PD in healthy volunteers, thus determining the first-in-human dose (FIH) and informing the efficacious dose in clinical settings. Model-derived results were considered to represent a rationale for the FIH dose selection.

Introduction underlines how low DC counts within the tumor microenvironment are likely to oppose the effectiveness of cancer immunotherapy (CIT) in cancer patients. In this context, the soluble growth factor FLT3L agonizes the fms-related tyrosine kinase receptor  (FLT3/CD135) with initiation of signalling cascades able to promote DC proliferation, differentiation, mobilization, and survival. Moreover, preclinical data seemed to favour a combination of FLT3L therapy with T-cell directed CIT including checkpoint inhibition.

In Materials and Methods, datasets used in the workflow of minimal PBPK/PD model development were detailed. PK model structure of FLT3L-Fc, GS-3583 and CDX-301 as well as PD model structure for DC expansion were also reported. Overview of model calibration, validation, and simulation included a series of  valuations including data from both cynomolgus monkey and human-derived scenarios.

Results describe the PK/PD model of the FLT3L-based treatments, the model calibration and validation in nonclinical species, the reproduction of clinical PK and PD from healthy volunteers receiving FLT3L (CDX-301), the model projection of clinical PK and PD following FLT3L-Fc treatment across dose level, and the rationale for FIH dose selection in healthy volunteers.

Discussion points out how the presented PBPK/PD model was able to integrate complex data from preclinical and clinical studies evaluating FLT3L-based therapies and to propose PK/PD profiles for FLT3L-Fc as well as clinical trial design and FIH dose selection are concerned. The performed clinical simulations allowed to select a safe starting dose of 700 mcg in a phase Ia clinical trial with an about 8-fold expansion of cDC1s in humans without exposure to sub-pharmacological doses. Furthermore, revised PK simulations between cynomolgus monkeys and humans were able to optimize receptor occupancy and cDC1 and total DC expansion for a given FLT3L-Fc dose. Tolerance to CDX-301 and GS-3583, clinical FLT3L-based molecules, was also considered for comparable exposures to the proposed FIH dose. It is concluded that future employments of the present model could include combination strategies to evaluate DC activation and anti-cancer effects in patients.

On the whole, the present study appears to have been well planned and carried out with rigorous methodologies, thus representing a valid contribution in the field of cancer immunotherapy. Lexicon, sentence fluency, “English style”, figures and their legends (6), and references are adequate.

Reviewer 3 Report

Comments and Suggestions for Authors

I enjoyed this paper as the studies are well conducted and it is sound and well written.

I would like additional validation information on the model's goodness of fit and its appropriateness for supporting clinical development.

Can you provide goodness-of-fit statistics, such as R-squared or other relevant metrics, for the developed minimal PBPK/PD model to demonstrate its fit to the pre-clinical and clinical data?

Have you conducted model validation using independent datasets or other techniques to assess its predictive performance? If so, could you share the results of this validation?

Could you provide the Schwarz criterion or the Akaike information criterion (AIC) values for the model to assess its relative goodness of fit and compare it to any alternative models you tried?

How robust is the model to variations in parameter values or assumptions? Have sensitivity analyses been performed to evaluate the impact of these variations on model predictions?

Overall, it is well done but I expect additional quantitative information on the model's performance and its suitability for supporting clinical development decisions.

Reviewer 4 Report

Comments and Suggestions for Authors

This manuscript establishes a minimal physiologically based pharmacokinetic/pharmacodynamic model with expansion-enhanced target-mediated drug disposition to integrate the mole-cule’s mechanism of action, as well as the complex preclinical and clinical PK/PD data, to support the preclinical-to-clinical translation of FLT3L-Fc. it has great significance for tumor therapy. However, some contents were not clearly stated, and English has many grammatical errors.

1 The FLT3L-Fc appeared the first time should use the whole name.

2 In the manuscript, there are too many keyword that should be shortened in line with the guidelines of the journal.

3 Line 32, eliminates

4 Line 32, because Anti-PD-L1/PD-1 appeared the first time, it should use the whole name.

5 Line 122-124, The list of studies includes 122 two internal preclinical studies in cynomolgus monkeys using FLT3L-Fc and 123 three clinical studies from other FLT3L-based therapies. Please add all references.

6 Line 127, please add references.

7 Line 144, although the model structure has been described in published references, reviewer think it should be explained here briefly.

8 Line 245, parameters of human PD

9 Line 245-247, the calibration of CDC1 used parameters of human PD. What indices did authors calculate? Please, displayed them and stated them in detail.

10 Line 248, what parameters did authors calculate to validate the model? Please, displayed them and stated them in detail.

Comments on the Quality of English Language

Extensive editing of English language required.

Round 2

Reviewer 4 Report

Comments and Suggestions for Authors

Reviewer did not see the responded letter.

Comments on the Quality of English Language

Moderate editing of English language required.

Round 3

Reviewer 4 Report

Comments and Suggestions for Authors

There is no effective reply for comment 1 and comment 10.

Comments on the Quality of English Language

Moderate editing of English language required.